# Study on Urbanization Sustainability of Xinjiang in China: Connotation, Indicators and Measurement

**DOI:** 10.3390/ijerph20032535

**Published:** 2023-01-31

**Authors:** Lei Kang, Siyou Xia

**Affiliations:** 1Key Laboratory of Regional Sustainable Development Modeling, Institute of Geographic Sciences and Natural Resources Research, Chinese Academy of Sciences, Beijing 100101, China; 2College of Resources and Environment, University of Chinese Academy of Sciences, Beijing 100049, China

**Keywords:** urbanization sustainability, evaluation, index, Xinjiang

## Abstract

BACKGROUND: Current research about sustainability evaluations in urbanization pays limited attention to certain areas of the world, thus potentially leading to an incomplete portrayal of the rich connotation of sustainable development. In fact, the existing evaluation criteria used by researchers in this field may not be generalizable due to regional variations. This study evaluated urbanization sustainability in Xinjiang Province (China) taking into account different perspectives, such as security and stability, social integration, economic vitality, happiness and livability, and ecological health. The aim was to develop an urbanization sustainability evaluation system, resulting in a new Index customized to regional characteristics and local development needs. METHODS: A spatial clustering analysis methodology was adopted to reveal the prominence of 15 issues in different areas of Xinjiang. RESULTS: Overall, the results showed low urbanization sustainability in Xinjiang, with significant intra-regional variability. The dimensions of security and stability scored the lowest in the newly developed Index, indicating specific aspects of weakness in Xinjiang’s urbanization sustainability. Social integration scored highly in the new index, implying that this aspect plays a supporting role in the urbanization sustainability of the region. Nevertheless, economic vitality scored low, representing a limitation for the region’s urbanization sustainability, as well as the happiness and livability dimensions. On the contrary, the parameter of ecological health scored high, despite spatial variances. Urbanization sustainability within each prefecture was further categorized as high, balanced, or low, revealing the main challenges faced by each prefecture during urbanization. CONCLUSIONS: The purpose of this study was to divert attention to the urbanization sustainability in different regions of the world, considering their particularity and diversity, thereby providing a research paradigm for scientific evaluation of urbanization sustainability.

## 1. Introduction

Urbanization is inevitable in economic and social development and is the path to modernization. China’s rapid urbanization has contributed significantly to the world’s total urbanization trend [1]. Since the 1990s, China has experienced rapid urbanization [2]. Consequently, many areas have endured the strong contradiction between increasing urbanization and demand for resources and environmental protection. This has led to the unprecedented speed and scale of urbanization, creating problems wherein urbanization of land is surpassing the mass movement of population groups, environmental pollution, weak integration of urban and rural areas, and poor development quality and efficiency [3,4,5,6,7]. In 2014, the “National New-Type Urbanization Plan (2014–2020)” was officially issued. Subsequently, the quality of urbanization has become the general focus [1]. In 2015, the United Nations adopted the 2030 Sustainable Development Goals (SDGs), urging world leaders to guide global sustainable development. The subsequent stage of urbanization planning targets the enhancement of traditional urbanization and aims to uphold the development needs of China’s social economy. In the process, it will ensure the integration of urban and rural areas, industry connectiveness, intensive and economical land use, ecological livability, and balanced development. The goal is to ensure alignment between capacity, speed, and quality [1,8]. The new urbanization plans are also consistent with the SDGs, as they are people-oriented measures that consider intergenerational interests for achieving coordination and sustainability in economic, social, resource, and environmental factors of urbanization. Achieving sustainable development is the only way urbanization can continue in China [9,10]; this will likely have a positive and far-reaching impact on sustainable development worldwide.

Urbanization involves a complex, dynamic, and multi-dimensional system of factors [11], and urbanization sustainability requires coordination across multiple dimensions. Therefore, evaluating urbanization sustainability must involve various indicators in the social, ecological, and cultural environment [12,13,14]. As urbanization intuitively implies the process through which cities develop and grow, it is also the underlying driving force. Most international studies on urbanization sustainability are based on the sustainable development of the corresponding city. Some evaluation methods have adopted a single comprehensive index, such as the following: Ecological Footprint, Environmental Sustainability Index, Human Development Index, City Development Index, green gross domestic product (green GDP), Genuine Progress Indicator, Living Planet Index, and Gross National Happiness Index [15,16,17]. These classic indices are widely used in urban sustainability research. Other evaluation systems have yielded findings based on multi-dimensional composite indices. Some scholars used the Sustainable Development for Energy, Water, and Environment Systems (SDEWES) Index to develop the best possible sustainable practices for Latin American cities for application through the framework of a solid SWEDES Index database of comparative analysis [18]. In the case study by Farzaneh Foroozesh et al. [19], which aimed to develop a practical framework to assess the development sustainability of Karaj, a new set of quantitative and qualitative criteria and sub-criteria tailored to the conditions of developing countries were employed. The main indicators in this process were SUD including Socio-Economic, Physical Characteristics of the land, Biological Environment, Buffer and Distances, Urban Infrastructure, Land Use, and Pollution Source criteria. Some scholars believe that cities can only maintain their prosperity and sustainable development goals when environmental and social objectives are fully integrated with economic goals. Therefore, they developed a system of indicators, including economy population, land and urban planning, energy, transportation, agriculture–livestock–fishery, industry, tourism, air pollution climate change, water resources and sea environment, solid waste, biodiversity, health, and education–research and technology factors to evaluate sustainability in urban areas and provide a dynamic tool for the management of environmental, social and economic information [20].

Regardless of whether the evaluation method used is based on the pressure–state–response model or the SDG framework, Chinese scholars have tended to incorporate resource, ecological, environmental, and socioeconomic factors to ensure a comprehensive evaluation. For instance, targeting the coordinated development of the economy, ecology, and society, Fang and Wei [21] developed an evaluation method for exploring the sustainable development capability of the Hexi region and revealed several patterns of regional variability. Referring to the SDG framework, Ma et al. [22] constructed an evaluation index system that included economic development, livelihood improvement, resource utilization, and environmental quality. They used the relative entropy method to assign combined weights to variables and emphasized the dynamic change and spatial distribution of urbanization sustainability in the Gansu Province. Ma et al. [23] developed an evaluation index system for urbanization sustainability that covers four aspects (i.e., economy, society, social resources, and environment), exploring urbanization sustainability in the Jilin Province from an intra- and inter-provincial perspective.

Abundant research is available on the sustainable urbanization assessment, and the construction of an index system is an important basis. Existing studies pay more attention to the universal application of sustainability evaluation indicators, and emphasize the dissemination of universal values under the concept of sustainable development. However, many distinct areas exist in the world which have their own unique historical development characteristics, reality, and future development demands. In addition, the urbanization sustainability presents multi-dimensional characteristics in different countries, regions, and cities. However, current research on sustainability evaluation of urbanization pays less attention to certain areas in the world, which will inevitably lead to an insufficient portrayal of the rich connotation of sustainable development, and will also limit the application of the concept of sustainable development in wider fields. Therefore, it is urgent to enrich research on the construction and evaluation of the sustainable development index system of urbanization in certain areas from a scientific perspective in order to better internalize the concept of sustainable development into the process of regional development. Overall, existing evaluation systems are diverse, and indicators usually cover multiple aspects of urbanization, such as society, economy, environment, and social infrastructure. Building an index system that incorporates all aspects of urbanization is conducive to understanding the complexities involved in sustainable urbanization. It is important to adopt generalizable indicators of sustainability. However, researchers must not overlook more specific, localized indicators, especially when examining urbanization at the regional level. Regions tend to vary in their urbanization stage, development, background, and goals, while specific regions may have unique regional characteristics and potential issues that impede sustainability. Hence, the overemphasis on the generalizability of indicators may neglect region-specific issues. Therefore, it is argued that any evaluation of urbanization sustainability applying a geographic demarcation approach must also include local geographical conditions. In addition to the generalizability and fit of the index system, other diverse, specific, and regionally representative indicators must be selected to reflect local urbanization goals fully. At the time of writing of this paper, no unified evaluation criteria have been agreed on for sustainable urbanization research. Thus, the development of a scientifically sound evaluation system for sustainable urbanization that reflects the characteristics and needs of specific regions is an important research topic.

For example, Xinjiang, located in the interior of Northwest China, has a unique geographical location and strategic position. A typical underdeveloped area, its urbanization has experienced a special evolutionary process and demonstrates a complex development status quo. Existing evaluation systems used in other regions may not be a good fit for such regions, as they fail to reflect the region’s true specific urbanization and sustainable development needs. A region-specific evaluation index system, developed with the incorporation of local factors, is a valuable addition to the relevant literature. Specifically, from the previous studies on Xinjiang urbanization, most of them set up index systems from population, economy, society, resources and environment, etc. [24,25,26,27]. The representation of Xinjiang’s urbanization development only remained at the traditional macro level, and the particularity of Xinjiang itself and the sustainability of urbanization development were not emphasized enough. Xinjiang’s urbanization displays both universality and particularity. Xinjiang’s security environment, ecological environment, resource endowment, and carrying function of urbanization are all unique. Its urbanization must take national defense security and social stability into account, adapt to the ecological environment, and integrate into ethnic unity and regional culture [28]. From this perspective, taking Xinjiang as an example, this study incorporates specific regional needs into the theoretical framework of sustainable urbanization. A five-dimension evaluation system was constructed, with 25 indicators covering aspects of security and stability, social integration, economic vitality, happiness and livability, and ecological health. The purpose was to quantitatively evaluate urbanization sustainability in Xinjiang and examine the issues faced by its different areas. Our research aims to enrich the case studies of urbanization sustainability, and strives to provide theoretical and practical paradigms for the scientific evaluation of urbanization sustainability in special regions.

## 2. Methodology and Research Data

### 2.1. Study Area

Xinjiang is situated in the hinterland of the Eurasian continent (Figure 1), and this region is a typical arid and semiarid area [29]. The region’s ecological environment is extremely fragile, as oases support production and living, resulting in specific ecological and environmental factors that influence sustainable urbanization considerably. As an underdeveloped region with a high population density of ethnic groups, urbanization in Xinjiang has not been an effortless process. Since the founding of the People’s Republic of China in 1960, Xinjiang has been developing rapidly. The region’s urban population increased from 13.2% of the total population in 1952 to 26.2% in 1960. However, the rate of urbanization declined shortly thereafter. In 1963, the urban population fell below 20%, declining to a historical low of only 17.4% in 1971 and remaining below 20% until 1975. Urbanization accelerated with China’s economic reform (i.e., reform and opening-up policy). Nonetheless, compared to the eastern regions, there remains a huge gap in the pace and quality of urbanization; such a gap appears to be growing, severely impeding the sustainability of regional economic development. Xinjiang has strategic significance and a unique set of attributes. Specifically, due to its geographical location and environmental factors both within and contiguous to Xinjiang, ensuring long-term stability is key to the continuous advancement of urbanization in the region. Moreover, multiple ethnic and religious groups in Xinjiang speak different languages and practice diverse cultures. Therefore, while maintaining national unity, urbanization must promote the integration of ethnicities and cultures, and Xinjiang has distinctive urbanization requirements.

### 2.2. Methodology and Data Sources

#### 2.2.1. Theoretical Foundations of the Sustainable Urbanization Index

Sustainable development comprises inclusive and coordinated development within a multi-dimensional system that includes the economy, ecology, and society. There are many indicators reflecting the above perspectives. However, if all the relevant indicators are considered and put into the system, it will lead to redundancy and the selection of indicators will become meaningless [24,30]. When compared to other regions, Xinjiang’s fragile ecosystem, underdeveloped economy, and unique social foundations demand an equally unique urbanization approach that incorporates the region’s practical needs with all its rich and complex nuances. Based on the region’s background, environment, and developmental needs, this study aimed to incorporate the region’s unique characteristics while acknowledging the general national requirements of sustainable development. This study explores sustainable urbanization based on five dimensions: security and stability, social integration, economic vitality, happiness and livability, and ecological health (Figure 2).

(1)Security and Stability: Prerequisites

Sustainable development emphasizes satisfaction of human needs [31], and safety is considered one of the most fundamental among these needs. Firstly, natural disasters are destructive and restrictive to urban development [32]. Xinjiang’s urban development is faced with fragile natural environment, and it is one of the regions with frequent occurrence of natural disasters in China [33]. Frequent natural disasters are bottlenecks in local urbanization [33,34]. Meanwhile, given the strategic mission of maintaining the long-term stability of oasis cities and towns, social security is crucial to the safety and coherence of the Xinjiang’s economic system. The urbanization development of Xinjiang needs to rely more on the high degree of urban public safety, stable employment environment, perfect public safety infrastructure construction and sufficient fund guarantee [35]. In this study, security and stability are represented as occurrence of natural disasters and the state of urban security. A society unconstrained by natural hazards and public security problems guarantees its citizens basic rights and interests, paving the way for efficient and sustainable urbanization. Indicators for this dimension include threats of natural disasters, public safety concerns, employment stability, urban construction and maintenance, and financial security.

(2)Social integration: Inherent Requirements

Xinjiang is a special region where multiple ethnic groups, languages and cultures coexist. Different ethnic groups differ greatly in religious beliefs and living habits. Cultural diversity brings new problems and challenges to Xinjiang’s urbanization development [36]. However, as an underdeveloped region, the weakest link in urbanization remains the gap between urban and rural development. Inevitably, this urban–rural gap has led to instability and deep-seated conflicts that have influenced urbanization [37]. As city–industry integration is central to sustainable urbanization, it has strengthened the foundations of Xinjiang’s industries; wide gaps in industrial development remain across prefectures, and the integration and coordinated progress of industries and urban infrastructure are yet to be achieved [38]. This lack of both “soft” and “hard” conditions has restricted the sustainability of urbanization. It can be seen that the dimension of social integration covers many aspects that need to be paid attention to in the sustainable development process of Xinjiang’s urbanization, including ethnic integration, urban–rural integration, industry–city integration, etc. Nonetheless, to achieve sustainable urbanization in Xinjiang, this study holds that the focus should be on integrating ethnic groups, ensuring urban–rural synergy and integration, and maintaining city–industry integration. The goal is to achieve efficient, sustainable and resilient urbanization.

(3)Economic Vitality: The Core Driving Force

Economic vitality refers to the ability and potential in the process of economic development [39]. Given the complex and ever-changing international and domestic environment, Xinjiang’s economy has now entered a “new normal” from high-speed to medium-to-high-speed growth. The extensive economic and social development caused by the high levels of input factors has become increasingly unsustainable, and capital scarcity has severely restricted the development of an export-oriented economy in the region [40]. The economic vitality of Xinjiang is mainly manifested in the ability of economic growth and capital introduction [41], which is the internal driving force of urbanization development [42]. There is an urgent need to transform the economic development model, promote the development of an advanced industry structure, enhance economic connectivity, and create a favorable environment for consumption and investment. The purpose is to stimulate the flow of labor, goods, and capital required to provide a continuous and enabling environment for urbanization.

(4)Happiness and Livability: The Goal

In recent years, Xinjiang began to focus on a new approach of accelerating people-centered urbanization [24], aiming to meet the basic physiological, safety, educational, developmental, and social security needs. Considerable attention has been paid to improving public services and municipal infrastructure to improve the quality of life. The quality of public services can be assessed based on the coverage of social welfare (such as education, medical care, and public cultural and sports facilities) and equal distribution of public services. Municipal infrastructure can be measured through road network density and urban green coverage in built-up areas. Overall, the promotion of these aspects is conducive to improving citizens’ happiness and livability of cities, thereby contributing toward sustainable urbanization.

(5)Ecological Health: An Important Guarantee

Xinjiang is located in the arid region of Northwest China, and its ecological environment is sensitive and fragile [43]. Urbanization in the region relies on oases ecosystem services. Effectively, the population carrying capacity of these water systems must be considered to ensure ecological security. Specifically, when evaluating the quality of urbanization, environmental costs such as water, land, and energy resources required for urban development must be considered, and reduction in environmental pollution must be included as an important criterion. Moreover, the new phase of urbanization in Xinjiang is focused on ensuring environmentally friendly and resource-efficient sustainable development. Therefore, it is imperative to improve pollution control and coordinate economic, social, and ecological benefits. As such, this study has selected several corresponding indicators to provide a reference for the coordinated development of sustainable urbanization and ecosystems in arid areas.

Based on the above theoretical foundations of sustainable urbanization and localized demands of Xinjiang, a five-dimension evaluation index system has been developed. The instrument includes 25 specific indicators measuring security and stability, social integration, economic vitality, happiness and livability, and ecological health (Figure 3).

#### 2.2.2. Entropy-Weighted TOPSIS

The entropy-weighted technique for order of preference by similarity to ideal solution (TOPSIS) method was used for developing the evaluation system [44,45]. The entropy-weighted TOPSIS method is a modification of the traditional TOPSIS method. The indicator weights are initially determined using the entropy method, and the best approximate ranking of solution of the TOPSIS method is applied to determine the optimum for the evaluated objects. The calculation steps are as follows:(1)Construct Matrix *X*

If there are *m* evaluation objects and *n* evaluation indicators, then matrix *X* can be constructed as follows:(1)$$X=(X_{ij}{)}_{m\times n}.$$

(2)Standardize Matrix *X* into Matrix *Y*

As the proposed evaluation system contains both positive and negative indicators, the standardized range of non-zero transformations were adopted to process the original data and yield the standardized matrix *Y*. The calculations for *Y_ij_* are shown in Formulas (2) and (3).

The standardization of positive indicators is defined in Formula (2).
(2)$$Y_{ij}=\frac{X_{ij}-X_{j\min}}{X_{j\max}-X_{j\min}}\times 0.99+0.01.$$

The standardization of negative indicators is outlined in Formula (3).
(3)$$Y_{ij}=\frac{X_{j\max}-X_{ij}}{X_{j\max}-X_{j\min}}\times 0.99+0.01,$$
where *Y_ij_*, *X_ij_*, *X_j_*_max_, and *X_j_*_min_ are the standardized, original, maximum, and minimum values of the *j*-th indicator of the *i*-th evaluation object, respectively.

(3)Determine the Weights for Evaluation Indicators

The two methods for determining the indicator weights are subjective and objective assignment methods. Among the objective assignment methods, the entropy weight method uses the inherent information of indicators to reflect its value of effect. Such an approach eliminates the noise of the subjective assignment method and is widely used in social and economic studies. In this study, the entropy- weight method was applied to obtain the weights of the indicators in the evaluation system. The calculation process was as follows:

① Calculate the proportion (*P_ij_*) of the *i*-th evaluation object under the *j*-th indicator based on the standardized *Y_ij_*:(4)$$P_{ij}=Y_{ij}/\sum_{i=1}^{m}Y_{ij}.$$

② Calculate the entropy (*e_j_*) of the *j*-th indicator:(5)$$e_{j}=-k\sum_{i=1}^{m}P_{ij}\ln P_{ij},\;\;\mathrm{where}\;k=1/\ln m.$$

③ Calculate the difference coefficient (*g_j_*) of the *j*-th indicator:(6)$$g_{j}=1-e_{j}.$$

④ Calculate the weight (*w_j_*) of the *j*-th indicator:(7)$$w_{j}=g_{j}/\sum_{j=1}^{n}g_{j}.$$

(4)Build the weighted standardized matrix *Z*
(8)Z=Y×W,
where $Z_{ij}=Y_{ij}\times W_{j}$.

(5)Determine the positive (*Z^+^*) and negative (*Z^−^*) ideal solutions:
(9)$$\begin{array}{l} Z^{+}=\left\{\max Z_{ij},j=1,2,\text{......}n\right\}=\left\{Z_{1}{}^{+},Z_{2}{}^{+},\ldots ,Z_{n}{}^{+}\right\}\\ Z^{-}=\left\{\min Z_{ij},j=1,2,\text{......}n\right\}=\left\{Z_{1}{}^{-},Z_{2}{}^{-},\ldots ,Z_{n}{}^{-}\right\} \end{array}.$$

(6)Calculate the Euclidean distance between each evaluation object and the positive/negative ideal solutions:(10)$${D_{i}}^{+}=\sqrt{\sum_{j=1}^{n}{\left(Z_{ij}-Z_{j}{}^{+}\right)}^{2}}\text{; }{D_{i}}^{-}=\sqrt{\sum_{j=1}^{n}{\left(Z_{ij}-Z_{j}{}^{-}\right)}^{2}}.$$

(7)Calculate the relative approximation distance (*C_i_*) of each evaluation object to the positive ideal solution:
(11)$$C_{i}=\frac{{D_{i}}^{-}}{{D_{i}}^{+}+{D_{i}}^{-}},$$
where 0 ≤ *C_i_* ≤ 1; the larger the *C_i_* value, the higher the ranking of the evaluation indicator and vice versa. In all the above formulas, *i* = 1, 2, …, m; while *j* = 1, 2, …, n.

#### 2.2.3. Regional Analysis

Xinjiang varies in natural and socioeconomic conditions and regional development advantages between its inner regions; hence, urbanization across its sub-regions is significantly different. As such, sub-regional analysis is conducive to the following: ensuring a diagnosis of the core issues required for formulating targeted policies; providing better guidance for local urbanization; allowing full play to the comparative advantages of each sub-region; promoting sustainable urbanization more effectively. Cluster analysis is a widely accepted method of grouping regions with similar characteristics. Therefore, the grouping analysis tool of ArcGIS 10.2 platform has been used in this study to group the sub-regions of Xinjiang and analyze their level of urbanization individually. Grouping analysis is a clustering method that uses unsupervised machine learning to determine natural groupings in data. Essentially, objects with strong similarity are clustered into one category, and those with broad differences are divided into distinct categories. Such an approach facilitates the analysis of similarities and differences between clusters. The basic principle of this method is to split data into a pre-defined number of groups based on various combinations of dimensions and compare the differences between the combinations to obtain a solution that ensures maximum within-group feature similarities and between-group feature differences. Feature similarity is a set of properties based on the analysis field parameters and can include spatial and spatio-temporal attributes. When a space or space–time constraint is defined, the algorithm uses a connected graph (minimum spanning tree) to detect natural groupings. When NO_SPATIAL_CONSTRAINT is specified, the Group Analysis tool adopts the K-means algorithm. In this study, NO_SPATIAL_CONSTRAINT was selected as the spatial constraint parameter, and the K-means algorithm was used for grouping. The objective was to group the administrative units based on the features of new urbanization to minimize the differences between the administrative units within each group.

### 2.3. Data Resources

The data used in this study were obtained mainly from Xinjiang Statistical Yearbook 2020 and China Urban Construction Statistical Yearbook 2019. At the same time, there were some data besides the yearbook. For example, POI data of public security organs, registered enterprises and cultural and sports facilities were mainly obtained from Baidu Map open platform in June 2020. The population data of all ethnic groups were obtained from the 2010 Census of Xinjiang Uygur Autonomous Region.

Some special indices were obtained by further processing and calculation. For example, the average distance of accessibility between rural settlements and cities was calculated by GIS spatial neighborhood analysis. Ethnic diversity was measured using the Shannon–Wiener diversity index, using the following formula:(12)$$HD=-\sum_{i=1}^{s}p_{i}\ln p_{i}=-\sum_{i=1}^{s}\frac{n_{i}}{N}\ln\frac{n_{i}}{N},$$
where *p_i_* represents the population proportion of the *i*-th ethnic group, and *n_i_* is the population of the *i*-th ethnic group. Further, *s* denotes the number of ethnic types, and *N* is the total population in a given geographical area. Broadly, a larger value in the diversity index indicates that a sizable number of ethnic groups live in the region, and the population of each ethnic group is large.

Additionally, owing to missing data, prefecture-level data were used for income and energy consumption of urban and rural residents; data for the proportion of days with air quality higher than level 2 were based on data covering major cities.

The data sources for all indicators are shown in Table 1.

## 3. Results

### 3.1. Status Quo and Characteristics of Urbanization Sustainability

#### 3.1.1. Overall Characteristics

As shown in Figure 4, the sustainable urbanization index and sub-dimension indices for prefectural-level municipalities were low. The urbanization sustainability varied sharply between cities and was markedly higher in northern Xinjiang than that in southern Xinjiang. A radar chart was used to illustrate the results for each dimension (Figure 5). The index of social integration scored the highest, followed by ecological health, whereas the indices of the other three dimensions remained low. Further, the indices for safety and stability, economic vitality, and happiness and livability were scattered, indicating that the quality of these dimensions varies widely across county-level cities. In particular, the index for safety and stability was the lowest, thus requiring urgent improvement focus. Specifically, the mean for the sustainable urbanization index was 0.147, and the indices of Urumqi, Karamay, Hami, Turpan, and Bortala Mongol Autonomous Prefecture, Kashgar Prefecture, and Ili Kazakh Autonomous Prefecture were greater than the mean. The Urumqi index scored the highest, with security and stability, economic vitality, and happiness and livability also ranking highest among all the sub-regions. The index of the Altay Prefecture was the lowest, and it also ranked the lowest in happiness and livability.

#### 3.1.2. Dimensional Characteristics

The results of analysis show that county-level cities with a high comprehensive index were distributed in northern Xinjiang, while those with a low security and stability index were mainly distributed in southern Xinjiang (Figure 6f). The spatial differentiation of the sustainability index was small, clearly indicating clustered distribution. The results also showed that 32.56% of county-level administrative units had a comprehensive index greater than the mean index for the Xinjiang region. Urumqi, Hotan, Alashankou, Kashgar, and Ghulja were the top five cities in terms of the comprehensive index results, with Urumqi ranked the highest (0.478). The comprehensive indices of 14 counties—Akqi, Luntai, Wuqia, Pishan, Baicheng, Hotan, Shule, Karakax, Wensu, Kuqa, Minfeng, Yengisar, Lop, and Jiashi—were less than 0.1, indicating low or poor quality of urbanization in these regions.

First, the index for security and stability scored the lowest, presenting the most severe issue that hampered sustainable urbanization. The mean for the security and stability index was 0.087—the lowest among the five dimensions. In terms of spatial distribution, the index for security and stability was higher in northern Xinjiang than in southern Xinjiang, thus presenting a general trend toward tilting from northeast to southwest. County-level units with a high security and stability index were scattered in northern Xinjiang, while those with a low security and stable index were distributed across large areas in southern Xinjiang (Figure 6a). Specifically, the indices for Urumqi, Alashankou, Toksun County, Hami, and Karamay ranked among the top five, while those for counties such as Yopurga, Karakax, Akto, Maralbexi, and Jiashi were the lowest.

Second, the index for social integration was high, which may have facilitated sustainable urbanization. The calculation results showed that the mean social integration index was 0.298, the highest among the five dimensions. In terms of spatial distribution, social integration showed heterogeneous spatial distribution, and counties with high and low integration presented a point-like scattered distribution (Figure 6b). Approximately 29.1% of the county-level administrative units had an index greater than the mean for the general Xinjiang region. Specifically, Tacheng and counties such as Makit, Poskam, Yarkant, Uqturpan, Jinghe, Shufu, Yumin, Qira, and Bohu had the highest index values (exceeding 0.5) and were ranked among the top ten. The indices of counties such as Toksun, Shanshan, Jeminay, Barkol Kazak Autonomous County, Fuyun, Keriya, Kuqa, Kargilik, Akto, Wuqia, Yopurga, Shule, Baicheng, and Jiashi were the lowest (less than 0.2).

Third, the index for economic vitality was generally low, which may have restricted sustainable urbanization. The results showed that the mean economic vitality index was 0.091, and the overall level was relatively low. Notable spatial variability was observed, and the spatial distribution of economic vitality overlapped slightly with security and stability. Specifically, the trend ran diagonally from the northeast to southwest, and county-level units with high economic vitality were scattered mainly in northern Xinjiang; those with low economic vitality were distributed across clusters in southern Xinjiang (Figure 6c). Only 20.93% of the county-level administrative units had an index above the regional mean, among which Kashgar, Hotan, Ghulja, Urumqi, Khorgas, and Alashankou had the highest indices (greater than 0.4). The indices for counties such as Maralbexi, Qapqal Xibe Autonomous County, Jiashi, Kargilik, and Karakax were the lowest (less than 0.025).

Fourth, the index for happiness and livability was relatively low, and spatial variance was prominent. The mean for the happiness and livability index was 0.115, and the indices of cities were generally low, with only 17 county-level units presenting an index greater than the regional mean. Figure 4d shows that county-level units with a high index showed a point-like distribution, while those with a low index showed a scattered–clustered distribution. Ghulja, Hotan, Kashgar, Kuytun, and Urumqi were ranked the top five cities in happiness and livability, with Ghulja and Hotan having the highest index values of 0.757 and 0.75, respectively. The indices of counties such as Maralbexi, Hotan, Kargilik, Lop, Karakax, and Yumin were the lowest (less than 0.05).

Finally, the index for ecological health was relatively high, and spatial variance was small, clearly indicating clustered distribution. The mean for the ecological health index was 0.208, ranking second among the five dimensions. Cities with high and low indices were concentrated in southern Xinjiang, presenting a notable clustered distribution (Figure 6e). Approximately 31.40% of county-level administrative units showed an index greater than the mean. Specifically, Alashankou, Shanshan, Hotan, Akto, and Aqsu were ranked among the top five in the ecological health index, with Alashankou showing the highest index value (0.527). Index values for counties such as Kuqa, Wensu, Shule, Pishan, Kashgar, Minfeng, Karakax, Lop, and Hotan were the lowest (less than 0.1).

### 3.2. Regional Characteristics of Urbanization Sustainability

In Section 3.1, urbanization sustainability in Xinjiang was analyzed from a comprehensive and dimension-based perspective. In this section, prefectural-level dimensions and comprehensive indices were calculated, and the Grouping Analysis of ArcGIS was used to categorize prefectures into different sub-categories. Specifically, the target number was three groups, and the indicators of five dimensions were all included in the analysis field. Further, the spatial constraint parameter was set at “NO SPATIAL CONSTRAINT,” and the K-means algorithm was used to minimize differences within each group.

The Xinjiang prefectures were divided into areas with high, balanced, and low-sustainability groups, based on the analysis results (Table 2). Urumqi was categorized as the only prefectural-level administrative unit belonging to the high-sustainability group. Prefectures in the balanced-sustainability group included Karamay, Kashgar, Hotan, Tacheng, Changji Hui Autonomous Prefecture, Bortala Mongol Autonomous Prefecture, and Ili Kazakh Autonomous Prefecture. Prefectures in the low-sustainability group included Turpan, Hami, Bayingolin Mongol Autonomous Prefecture, Aksu, Altay, and Kizilsu Kyrgyz Autonomous Prefecture. As the capital of the Xinjiang Uygur Autonomous Region, Urumqi is in the lead position within the urban system. Urumqi ranks the highest for many economic indicators, and the social environment remains generally stable. The local government is focused on the simultaneous development of the economy and urban livability. The results show that the indices for economic vitality, safety and stability, and happiness and livability of Urumqi were 0.4526, 0.6282, and 0.3190, respectively, which were significantly higher than those of other prefectures. However, only the ecological health index was relatively low. Notably, the indices for economic vitality, safety and stability, and happiness and livability for the balanced-sustainability group were lower than those for the high-sustainability group. Nonetheless, the indices for social integration (0.3318) and ecological health (0.1947) were greater compared to those for the high-sustainability group. Moreover, except for the ecological health index, the four dimensions of the index in the balanced-sustainability group were greater than those of the low-sustainability group. The low-sustainability group had a high ecological health index (0.2742), but the other four dimensions were low. Prefectures in this group are located in southern Xinjiang with an underdeveloped economy and weak self-development capabilities, which perhaps significantly restricts sustainable urbanization efforts.

## 4. Discussion

### 4.1. Dimensional Analysis of Urbanization Sustainability in Xinjiang

Promoting sustainable urbanization is an important process for achieving leapfrog development in Xinjiang. This study determined that economic vitality was largely low, which is the major constraint impeding sustainable urbanization in Xinjiang. Urban development is inseparable from economic development, and an underdeveloped economy directly limits the financial resources available for urban development. Further, compared to other regions of China, Xinjiang has fewer cities and towns, and most of them are small and scattered [46]. Lack of economic development directly affects the agglomeration of industries and population, which, in turn, affects the expansion and development of small and medium-sized cities and towns. As small and medium-sized cities and towns are important centers for new urbanization [47], economic limitations in these cities and towns significantly hamper improvement in urbanization quality. This study determined that the security and stability index was the lowest among all dimensions. Combining the indicators selected to measure this dimension, the reasons could be inferred. First, in terms of natural hazards, Xinjiang is an area frequently affected by natural disasters [24], which occur over a widely distributed area and tend to be diverse and intense [48]. Such disasters restrict land use development and are detrimental to the construction of economic infrastructure. In terms of social stability and public security, although the regional situation is generally stable and improving, ethnic and potential security problems at the frontier make the region vulnerable to a high incidence of emergencies and weak social governance. These factors have posed challenges to the advancement of sustainable urbanization. In terms of employment stability, both secondary and tertiary industries are increasingly capable of providing jobs, and employment continues to expand. Nevertheless, due to the surplus agricultural population and weak economic foundations, the service industry continues to lag the large labor force, which has resulted in a high unemployment rate. Furthermore, due to the changing economic structure, regional imbalance in development, and misalignment between labor quality and market demand, people frequently lose their jobs, and employment stability is overall weak [49]. All these factors result in security and stability being the most critical issue in the advancement of new urbanization. Additionally, the overall happiness and livability index was low. In this study, this index was mainly composed of indicators based on the supply of public services. Prior studies have shown that the provision of public services is seriously poor in Xinjiang [50], while the lack of economic development has a direct and negative impact on investment in public services. Thus, these issues were not conducive to creating a social environment that supports sustainable urbanization. The results also show that Xinjiang has a strong social integration environment, which may be attributable to the continuous advancement of inclusive, integrated, and participatory development, alongside the pursuit for more equitable and sustainable development. Broadly, there is mutual respect and inclusiveness between various ethnic groups in Xinjiang. There is also continuous enhancement of urban–rural connectivity [51] and the integration of varied social identities of the migrant populations [52]. These factors have undoubtedly played a positive role in the development of sustainable urbanization. The high overall ecological health index was closely associated with the continuous emphasis on pollution control, green and low carbon development, and ecological environmental protection and construction in recent years [53,54]. Notably, these measures have improved the quality of the ecological environment in Xinjiang, and such improvements serve as foundations for future sustainable urbanization.

### 4.2. Diagnosis of Problems by Sub-Region

This study applied spatial analysis (group analysis) to categorize the prefectures into distinct groups to distinguish the sustainability of urbanization and the main problems of each sub-regional group. Specifically, first, only Urumqi was categorized as belonging to the high-sustainability group. Comparing the dimensional indices with other groups showed that ecological health was Urumqi’s main shortcoming. The oasis system of Urumqi implies inherent ecological fragility. Further, as an extra-large city evolving and experiencing rapid development, Urumqi saw previous urbanization policies focusing largely on economic growth and industrialization, causing enormous stress on the ecological environment. For instance, although water resources were scarce in the region, there was overexploitation of water. Industrial gas emissions and discharge of wastewater and residual waste were heavy, and air pollution became severe. All these factors collectively exacerbated the importance of urban ecological and environmental problems [55], thereby having a negative impact on urbanization sustainability. Second, for areas with balanced urbanization sustainability, there were no apparent shortcomings in the varied dimensions, and the index for social integration was robust. These factors supported the promotion of sustainable urbanization. However, compared to the high-sustainability group, economic vitality, safety and stability, and happiness and livability could be further improved, and ecological health issues should not be ignored. This group covered most of the prefectures in northern Xinjiang and the Kashgar and Hotan Prefectures in southern Xinjiang. These areas, which developed mostly around oases where productivity is high, developed modern agriculture and industrial infrastructure and contributed significantly to the economy than other areas in Xinjiang. Nevertheless, in terms of economic vitality, although the northern prefectures have numerous industrial enterprises, emphasis on agriculture, forestry, animal husbandry, and fisheries was sizable. Bortala Mongol Autonomous Prefecture and Ili Kazakh Autonomous Prefecture relied largely on primary industry output, the industrial economy was dominated by resource development, and the level of industrial structure was at a relatively poor level. Kashgar and Hotan Prefectures in southern Xinjiang were representative of areas with weak economic foundations, specifically areas that lacked infrastructure and public service facilities and where the level of per capita public health resources was low. Further, the harmless treatment of urban waste and wastewater was underdeveloped. In the absence of the abovementioned basic driving forces, the indices of security and stability and happiness and livability lagged the high-sustainability group. Many prefectures and cities in this group—Karamay, Changji Hui Autonomous Prefecture, and Ili Kazakh Autonomous Prefecture—serve as the fulcrum of the economic belt on the northern slope of the Tianshan Mountains. With rapid economic development and the growth of the local population, the ecological environment continues to remain under pressure, which, in turn, has restricted sustainable urbanization advancements. Third, for areas with low urbanization sustainability, ecological health was the only indicator that scored high, and the endogenous driving force for sustainable urbanization was insufficient. Most prefectures in this group were economically underdeveloped, with low city and town density. Agricultural populations continued to dominate many towns, and the development of secondary and tertiary industries lagged. Consequently, infrastructure construction was limited, and the urbanization process was slow. These prefectures must improve their respective independent capabilities inclusively to promote high-quality and sustainable urbanization efforts.

### 4.3. Political Implications

The study findings show that the characteristics of Xinjiang’s prefectures are highly variable in terms of urbanization. Further, the degree of urbanization was determined to be different in each sub-region, and the problems faced by each prefecture were considerably distinct. In such a context, a unified advancement model may not be applicable. Based on the qualities of the identified groups, the following sustainable urbanization pathways are proposed.

First, as the only area with high sustainable urbanization, Urumqi must focus on ecological health while steadily improving its urbanization quality. Specifically, it is suggested that the local government must strive to improve green development, properly balance and coordinate socioeconomic development, and ensure ecological and environmental protection. In the process, it should vigorously promote the governance and protection of the ecological environment, focus on green development strategies, and shift emphasis away from the more traditional views of urbanization being entirely a factor of urban construction [56]. It is also recommended to actively promote green production and sustainable lifestyles, continuously increasing the efficiency of energy and resource utilization and further improving the living environment.

Second, it is recommended that prefectures in the balanced-sustainability group construct new oases based on the principles of sustainable development while carefully coordinating existing oases resources and vigorously developing small and medium-sized cities and towns. The focus must be on improving industry structure—continuously promoting and upgrading traditional industries and enhancing industrial development capabilities to support new urbanization efforts. Further, the local government must engage actively with the principles of new urbanization, strive to improve citizens’ quality of life, accelerate the construction of key infrastructure in urban and rural areas, and continue to improve the living environment. Moreover, the ideal of ecological civilization must be fully integrated into the new urbanization process, with a strong focus on resource conservation and development of environmentally friendly towns. Additionally, policies and national systems must be used to impose rigid constraints, energy consumption must be reduced to alleviate the high pressure of economic development on the ecological environment, and sustainable urbanization must be promoted consistently.

Third, prefectures in the low-sustainability group lack self-development capabilities across several key dimensions and areas. Therefore, it is necessary to increase investment in developing such capabilities by strengthening existing infrastructure. Further, additional support must be provided to secondary and tertiary industries, and leading industries such as textiles and garment manufacturing must be promoted and expanded. Moreover, there should be rigorous production of petroleum, natural gas, and chemicals, alongside processing agricultural and sideline products and providing more employment opportunities for migrant workers. Additionally, prefectures must allow full play to the health benefits of low-carbon ecological systems and focus on improving the quality of urban systems to attract construction investment.

### 4.4. Strengths and Limitations

Urbanization sustainability ultimately needs to achieve coordinated development of population, economy, society, and ecology, which is a reflection of comprehensive ability. However, different regions have different personalized characteristics in geographical environment, historical conditions, urbanization evolution process and other aspects, so the evaluation indicator selection of urbanization sustainability and the obstacles to urbanization sustainability are inevitably different. It is the goal of this study to construct an efficient and scientific index system, according to local conditions, to evaluate the urbanization sustainability in a specific region and guide the region to take a healthy and orderly path of urbanization. In this paper, Xinjiang, one of the typical special regions in China and even the world, was selected as the study area. The natural background, historical track of development and socio-economic conditions of its urbanization determine the particularity, vulnerability, and development capacity gap of the urbanization in this region in terms of ecological environment, social culture and economic construction. Sustainable development is a global issue, and the urbanization sustainability of such special region is particularly important for the realization of the global sustainable development goals. Current studies on urbanization evaluation of Xinjiang mostly focused on population, land, economy, society and other basic aspects. Compared with previous studies on urbanization development evaluation of many other regions, there were no obvious new ideas, and the particularity and development demands of Xinjiang were not adequately reflected. In this paper, the urbanization sustainability level of Xinjiang was evaluated. While reflecting the comprehensive characteristics of sustainable development, some relatively unique practical demands of Xinjiang in the aspects of security, stability and social integrated development were included into the evaluation index system, which objectively reflected the uniqueness of the region in the process of urbanization promotion, and strived to achieve the integration of scientific logic and regional reality. This paper makes useful exploration for comprehensive evaluation of sustainable development of special regional urbanization.

Of course, the construction of any evaluation index system has certain subjectivity. The urbanization sustainability itself is a multi-dimensional and complex process with rich connotations, and it involves multiple and complex indicators. Different scholars may have different understandings of the indicators, and it is difficult to form universally accepted standards. This study aims to focus on the key problems and contradictions that Xinjiang is facing in the practice of urbanization development, and take into full consideration the main goals and challenges in the process of its social and economic development as much as possible to select evaluation indicators. However, due to the limitations of the author’s cognitive level and the availability of statistical data, the selection of indicators is not refined enough and the coverage of indicators is not comprehensive. Moreover, the development of regional urbanization is closely related to the social and economic reality, and its connotation will continue to develop and change. In the follow-up study, we will continue to refine the evaluation indicators and improve data collection in order to reflect the dynamic changes and development of Xinjiang’s urbanization process as scientifically and comprehensively as possible.

## 5. Conclusions

Xinjiang’s urbanization has been a unique evolutionary and complex developmental process. As it is an underdeveloped region in China, evaluation systems relevant to other regions may not be directly applicable. In the specific context of Xinjiang, this study highlighted the distinctiveness of Xinjiang’s development while abiding by the core principles of sustainable development. The aim was to construct an index that quantitatively evaluated urbanization sustainability from the perspective of five dimensions—security and stability, social integration, economic vitality, happiness and livability, and ecological health. Spatial clustering analysis was adopted to reveal the main problems faced by different sub-regions. The findings served as a decision-making basis for formulating a sustainable urbanization policy in Xinjiang.

Sustainable urbanization of many regions in the world is uniquely shaped by their distinct natural environment, diverse economic and social development foundations, development advantages and disadvantages, and differentiated development demands. The concept of sustainable development advocates inclusiveness. More attention must be paid to such regions and an urbanization sustainability evaluation index system that meets the specific development needs of the region must be built. Such a system would demonstrate the rich connotation of sustainable development from different angles, and form regional urbanization sustainable development models with different emphasis. Based on the requirements of sustainable development and focusing on the urbanization realities, we evaluated the urbanization sustainability level of Xinjiang. While reflecting the comprehensive characteristics of sustainable development, the practical demands of Xinjiang in the aspects of security, stability and integrated development were included into the evaluation index system. We strived to achieve the integration of scientific logic and regional reality and made useful exploration for comprehensive evaluation of urbanization sustainability of the special region.

Sustainable development is a concept and a great practical activity in human history. The urbanization sustainability belongs to this category. Thus, it is a theoretical as well as a practical problem with a realistic basis. Relevant research is necessary to prevent the disconnect between theory and practice. In order to achieve the sustainable development goals, we should consider all types of regions from an inclusive perspective. While pursuing the universal values advocated by sustainable development, the particularity, differences, and diversity of regions must be considered. Only in this way can we better avoid the disconnect between theory and practice, and lead the concept of sustainable development to better guide the development of human society.

## Figures and Tables

**Figure 1 ijerph-20-02535-f001:**
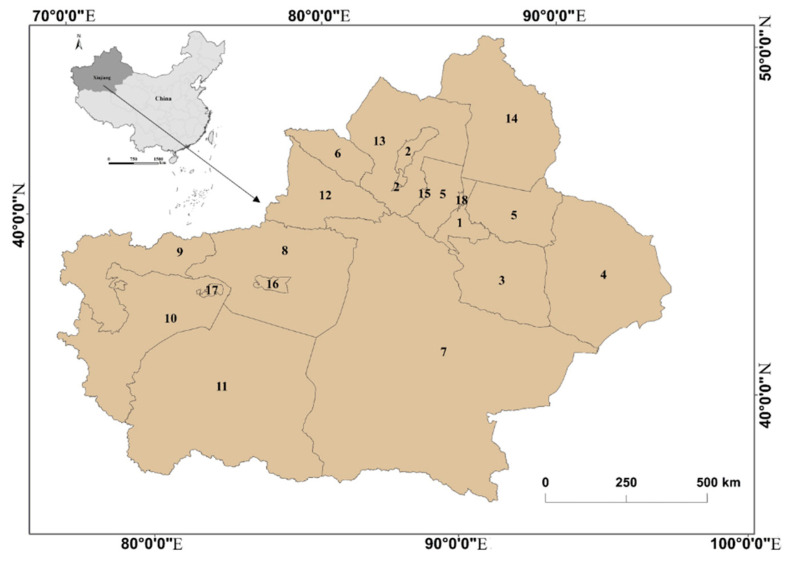
Location map of the study area. Note: Codes 1–18 in the map represent Urumqi, Karamay, Turpan, Hami, Changji Hui Autonomous Prefecture, Bortala Mongol Autonomous Prefecture, Bayingolin Mongol Autonomous Prefecture, Aksu, Kizilsu Kyrgyz Autonomous Prefecture, Kashgar, Hotan, Ili Kazakh Autonomous Prefecture, Tacheng, Altay, Shihezi city, Aral City, Tumxuk, and Wujiaqu City, respectively.

**Figure 2 ijerph-20-02535-f002:**
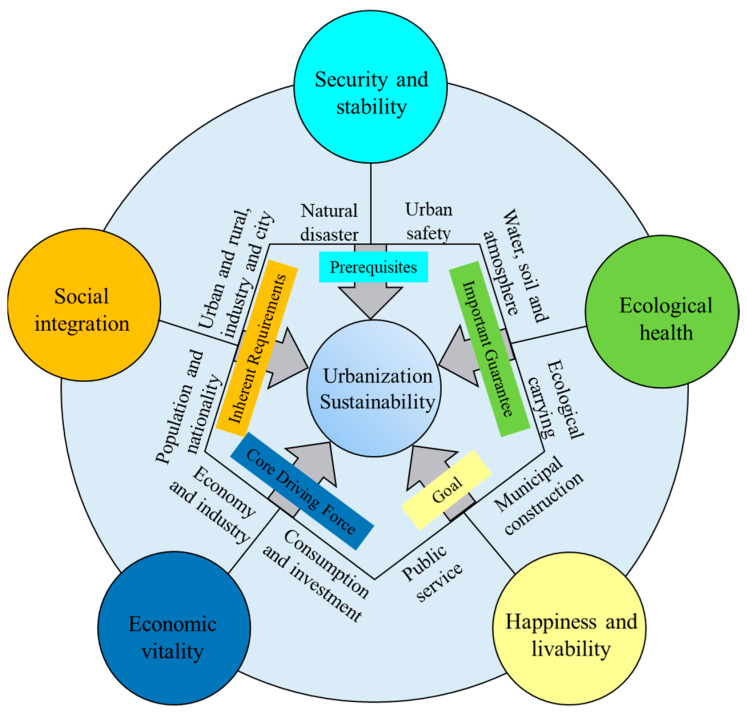
Theoretical connotation of urbanization sustainability of Xinjiang.

**Figure 3 ijerph-20-02535-f003:**
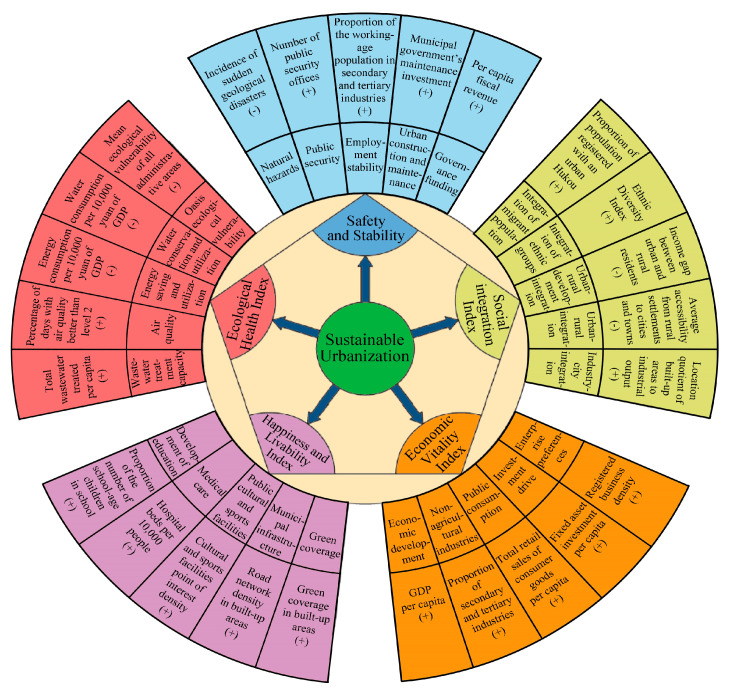
The evaluation index system of Xinjiang’s urbanization sustainability.

**Figure 4 ijerph-20-02535-f004:**
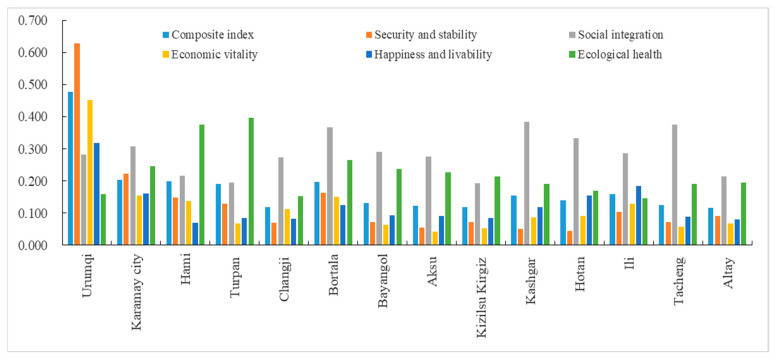
Composite index and sub-dimension index of urbanization sustainability in Xinjiang.

**Figure 5 ijerph-20-02535-f005:**
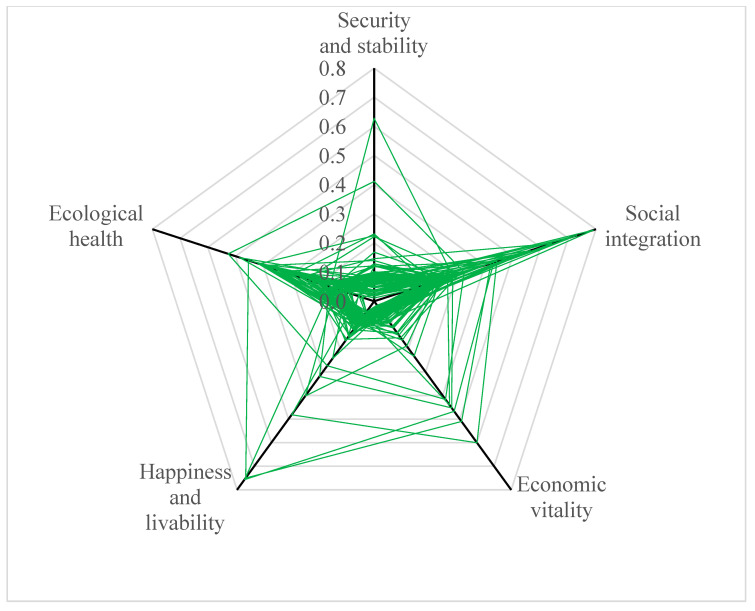
Radar chart of urbanization sustainability by dimension.

**Figure 6 ijerph-20-02535-f006:**
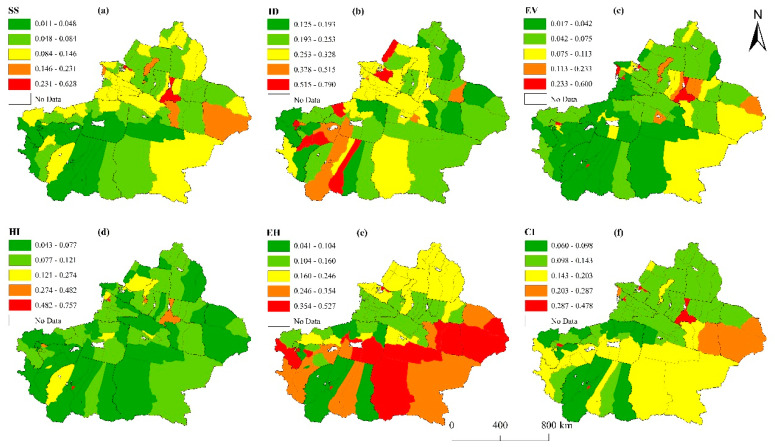
Illustration of the spatial distribution of the dimensional and comprehensive indices for each county (city). Note: Figures (**a**–**f**) represent the grading diagram of the evaluation results from five dimensions of Safety and Stability, Social Integration, Economic Vitality, Happiness and Livability, Ecological Health, respectively.

**Table 1 ijerph-20-02535-t001:** Summary of evaluation index of urbanization sustainability in Xinjiang.

Dimension	Indicator	Meaning	Sources
Safety and Stability (SS)	Natural disaster stress degree	Level of sudden geological disasters in Xinjiang−	Assessment map of resources and environment carrying capacity in Xinjiang (Resource and Environment Science and Data Center, CAS)
	Urban public safety level	Number of public security organs+	POI data, Baidu Map open platform in June 2020
	Urban employment stability	The proportion of the population working in the secondary and tertiary industries+	County Statistical Yearbook
	Urban construction and maintenance level	Municipal maintenance investment+	City/County Construction Statistical Yearbook
	Management fund security degree	Per capita fiscal revenue+	County Statistical Yearbook
Social Integration Index (SI)	Immigrant integration degree	Urbanization rate of registered population+	Xinjiang Statistical Yearbook
	Diversity of ethnic integration	Ethnic Diversity Index+	Ethnic Volume of the Sixth Population Census
	Integration degree of urban and rural development	The income gap between urban and rural residents−	Statistical Yearbook of Xinjiang (Prefecture level instead)
	Urban–rural integration	Mean accessibility of rural settlements to towns−	Calculated by GIS spatial neighborhood analysis
	City and industry development integration degree	The ratio of built-up area proportion to industrial output proportion+	County Statistical Yearbook and City/County Construction Statistical Yearbook
Economic Vitality Index (EV)	Economic development level	Per capita GDP+	County Statistical Yearbook
	Non-agricultural industry vitality	The proportion of secondary and tertiary industries+	County Statistical Yearbook
	Public consumption vitality	Per capita retail sales of consumer goods+	County Statistical Yearbook
	Investment drives level	Per capita fixed asset investment+	County Statistical Yearbook
	Enterprise preference	Registered enterprise density+	Enterprise POI database
Happiness and Livability Index (HL)	Education development level	The proportion of students in school+	County Statistical Yearbook
	Medical security level	Number of beds per ten thousand people+	County Statistical Yearbook
	Culture and sports sharing level	Cultural and sports facilities POI density+	POI data, Baidu Map open platform in June 2020
	Municipal construction level	Road network density in built-up area+	City/County Construction Statistical Yearbook
	Green space cover level	Green coverage rate of built-up area+	City/County Construction Statistical Yearbook
Ecological Health Index (EH)	Sewage treatment capacity	Total sewage treatment per capita+	City/County Construction Statistical Yearbook
	Air quality level	The proportion of days with better air quality than Grade II+	City/County Construction Statistical Yearbook
	Energy conservation level	Energy consumption per ten-thousand-yuan GDP−	City/County Construction Statistical Yearbook
	Water resource conservation level	Water consumption of ten-thousand-yuan GDP−	City/County Construction Statistical Yearbook
	Oasis ecological vulnerability	Average value of ecological vulnerability of administrative region−	Resource and Environment Science and Data Center, CAS

**Table 2 ijerph-20-02535-t002:** Statistics of urbanization sustainability zoning in Xinjiang.

Zone	Variable	Mean Value	Standard Deviation	Minimum	Maximum	Region
High-sustainability areas	Economic Vitality	0.4526	0.0000	0.4526	0.4526	Urumqi
	Safety and Stability	0.6282	0.0000	0.6282	0.6282	
	Happiness and Livability	0.3190	0.0000	0.3190	0.3190	
	Social integration	0.2829	0.0000	0.2829	0.2829	
	Ecological Health	0.1599	0.0000	0.1599	0.1599	
Balanced-sustainability areas	Economic Vitality	0.1113	0.0334	0.0564	0.1551	Karamay, Kashgar, Hotan, Tacheng, Changji Hui Autonomous Prefecture, Bortala Mongol Autonomous Prefecture, and Ili Kazakh Autonomous Prefecture
	Safety and Stability	0.1033	0.0610	0.0435	0.2219	
	Happiness and Livability	0.1306	0.0347	0.0835	0.1836	
	Social integration	0.3318	0.0413	0.2726	0.3836	
	Ecological Health	0.1947	0.0421	0.1472	0.2662	
Low-sustainability areas	Economic Vitality	0.0718	0.0309	0.0426	0.1379	Turpan, Hami, Bayingolin Mongol Autonomous Prefecture, Aksu, Altay, and Kizilsu Kyrgyz Autonomous Prefecture
	Safety and Stability	0.0945	0.0333	0.0554	0.1485	
	Happiness and Livability	0.0836	0.0077	0.0691	0.0922	
	Social integration	0.2308	0.0382	0.1926	0.2897	
	Ecological Health	0.2742	0.0804	0.1960	0.3975	

## Data Availability

The data sources in this study were specified in the paper, and corresponding index data can be obtained according to relevant data sources. No new data were created.

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
