# Peer review of "Study on Urbanization Sustainability of Xinjiang in China: Connotation, Indicators and Measurement"

_ijerph, 2023, doi:10.3390/ijerph20032535_

Round 1

Reviewer 1 Report

This paper focused on sustainable development which is one of the international hot topics, studied the urbanization development of special areas, established a multi-dimensional evaluation index system, and revealed the main problems faced by different regions in the process of sustainable development of urbanization by using certain spatial analysis methods on the basis of quantitative measurement. The topic shows certain leading characteristics. Different regions tend to vary in their urbanization stage, development, background, and goals, while specific regions may have unique regional characteristics and potential issues that impede sustainability. This paper emphasized the combination of general index and characteristic index in the construction process of index system, and highlighted the scientific nature of treating problems "according to local conditions" from the perspective of geography, which has certain enlightening significance for the research on issues related to the sustainability of urbanization in specific regions.

In addition, suggestions on promoting sustainable development of urbanization were given for different types of areas. Overall, this paper presents a very meaningful work and has enriched the case studies of sustainable development of urbanization. At the same time, the authors have an objective understanding of the strengths and limitations of the paper itself. It can be further improved from the following aspects.

1. This paper introduced the reasons for choosing Xinjiang as the study area of urbanization sustainability evaluation. Then, what researches have scholars carried out on the urbanization development of Xinjiang? What are the shortcomings? The paper can add some discussion on this aspect.

2. In the introduction part, the particularity of Xinjiang region can be appropriately added to highlight the significance of choosing Xinjiang as the research area of this paper.

3. The theoretical connotation of Xinjiang’s urbanization sustainable development index system is actually the basis for the selection of indicators below. At present, the interpretation of the five dimensions is more general, which can be further deepened and supplemented with some supporting arguments.

4. This paper established urbanization sustainability evaluation index system from five dimensions, and explained its applicability to the special region, Xinjiang. Could the authors further clarify which indicators of this index system are general and which indicators focused on reflecting regional characteristics? What is the superiority of this index system compared with the current evaluation research on urbanization development? Please further summarize and supplement.

5. If the comparison with existing studies can be further increased, it is meaningful to demonstrate the scientificity of the results in this paper.

6. The promotion path and suggestions of sustainable urbanization in different prefectures of Xinjiang can be simplified to some extent.

7. At present, the discussion part of this paper has been very rich. If some references can be added and supplemented, it will be conducive to better supporting the relevant research conclusions of this study. At the same time, the discussion section of the article is a little bit too long. It is suggested to further simplify and condense, so that readers can better capture the author's core ideas.

Author Response

Many thanks for the insightful comments and suggestions concerning our manuscript (ID: ijerph-2138243). We have considered your comments and have incorporated almost all your suggestions in the revised version of our paper. We also have a detailed response to each comment in the below. In the paper, the changes made are clearly highlighted using the "Track Changes" function. We feel that by revising our paper according to your constructive suggestions, the paper has been greatly improved.

The following are the answers and revisions.We have made in response to the reviewers' questions and suggestions on an item by item basis.

Reply to Referee 1

Specific Comments 1:

This paper introduced the reasons for choosing Xinjiang as the study area of urbanization sustainability evaluation. Then, what researches have scholars carried out on the urbanization development of Xinjiang? What are the shortcomings? The paper can add some discussion on this aspect.

Response to comment 1:

Thanks a lot for the helpful suggestions. We have revised and added related content according to your comment. We pointed that most of previous researches set up index systems from population, economy, society, resources and environment, etc. The representation of Xinjiang’s urbanization development only stayed at the traditional macro level, and the particularity of Xinjiang itself and the sustainability of urbanization development were not emphasized enough, which is the most obvious deficiency. Please see the added content in line 142-147.

Specific Comments 2:

In the introduction part, the particularity of Xinjiang region can be appropriately added to highlight the significance of choosing Xinjiang as the research area of this paper.

Response to comment 2:

Thanks for your positive and valuable comments.

In fact, we have mentioned the particularity of Xinjiang's urbanization development in the section 2.1 “Study area”. But it's really not enough in the introduction. By further summarizing the existing research results, we have added and revealed at the end of the introduction, that Xinjiang's urbanization has both universality and particularity. Xinjiang's security environment, ecological environment, resource endowment, and carrying function of urbanization are all unique. Its urbanization must take national defense security and social stability into account, adapt to the ecological environment, and integrate into ethnic unity and regional culture. Please check the revised version in Line 148 -152.

Specific Comments 3:

The theoretical connotation of Xinjiang’s urbanization sustainable development index system is actually the basis for the selection of indicators below. At present, the interpretation of the five dimensions is more general, which can be further deepened and supplemented with some supporting arguments.

Response to comment 3:

Thanks for your valuable comments. We have further explained the connotation of key dimensions, and supplemented with more supporting references. Please check the revised version in Section 2.2.1.

Specific Comments 4:

This paper established urbanization sustainability evaluation index system from five dimensions, and explained its applicability to the special region, Xinjiang. Could the authors further clarify which indicators of this index system are general and which indicators focused on reflecting regional characteristics? What is the superiority of this index system compared with the current evaluation research on urbanization development? Please further summarize and supplement.

Response to comment 4:

Thanks a lot for your valuable comments.

In fact, we have emphasized the superiority of the index system in the discussion part. To be specific, we pointed out that “in this paper, Xinjiang, one of the typical special regions in China and even the world, was selected as the study area. The natural background, historical track of develop-ment and socio-economic conditions of its urbanization determine the particularity, vulnerability, and development capacity gap of the urbanization in this region in terms of ecological environment, social culture and economic construction. Sustainable development is a global issue, and the urbanization sustainability of such special regional is particularly important for the realization of the global sustainable development goals.” in Line 663-669.We also mentioned that “some relatively unique the practical demands of Xinjiang in the aspects of security, stability and social integrated development were included into the evaluation index system, which objectively reflected the uniqueness of the region in the process of urbanization promotion” in Line 675-678, which presented that security, stability and social integrated development belongs to indicators focusing on regional particularities.

According to your suggestion, we have further supplemented that “Current studies on urbanization evaluation of Xinjiang mostly focused on population, land, economy, society and other basic aspects. Compared with previous studies on urbanization development evaluation of many other regions, there were no obvious new ideas, and the particularity and development demands of Xinjiang were not adequately reflected.” Please check the revised version in Line 670-673.

Specific Comments 5:

If the comparison with existing studies can be further increased, it is meaningful to demonstrate the scientificity of the results in this paper.

Response to comment 5:

Thanks a lot for your valuable comments. At present, we haven't added much to the existing research for comparison. The reasons are as follows: Based on the improvement and development of the existing urbanization evaluation index system, this study built a set of urbanization sustainability evaluation index system suitable for special areas like Xinjiang. However, current studies on urbanization evaluation of Xinjiang mostly focused on population, land, economy, society and other basic aspects. Compared with previous studies on urbanization development evaluation of many other regions, there were no obvious new ideas, and the particularity and development demands of Xinjiang were not adequately reflected. This is also related to the differences in the key issues of urbanization development that different researchers pay attention to. Therefore, it is not easy for us to make a comprehensive and objective comparison between relevant studies with the limited words However, inspired by the suggestions of the reviewer, in order to show the scientificity of the results, a certain number of references have been added in the discussion part, and relevant views are quoted to support the evaluation results of this paper, so as to ensure the rationality and scientificity of our research results.

Specific Comments 6:

The promotion path and suggestions of sustainable urbanization in different prefectures of Xinjiang can be simplified to some extent.

Response to comment 6:

Thanks a lot for your suggestions. According to the comment, we have made a large degree of deletion in the section of Political Implications. Please check the revised version.

Specific Comments 7:

At present, the discussion part of this paper has been very rich. If some references can be added and supplemented, it will be conducive to better supporting the relevant research conclusions of this study. At the same time, the discussion section of the article is a little bit too long. It is suggested to further simplify and condense, so that readers can better capture the author's core ideas.

Response to comment 7:

Thanks a lot for your suggestions. According to the comments, we have added a certain number of references in the discussion part, and deleted part of the expression to ensure that the article is concise. Please check the revised version.

Reviewer 2 Report

This study incorporates specific regional needs into the theoretical framework of sustainable urbanization,taking Xinjiang as an example. A five-dimension evaluation system was constructed, with 25 indicators covering aspects of security and stability, social integration, economic vitality, happiness and livability, and ecological health. The research quantitatively evaluated urbanization sustainability in Xinjiang and examine the issues faced by its different areas. In particular, the study developed the evaluation index system by combining the general indicators and regionally representative indicators to reflect local urbanization goals fully, because specific regions may have unique regional characteristics and potential issues that impede sustainability. This paper diverted attention to the urbanization sustainability in distinct regions in the world, considering their particularity and diversity, thereby providing a research paradigm for scientific evaluation of urbanization sustainability in distinct regions, which could help us better avoid the disconnect between theory and practice, and lead the concept of sustainable development to better guide the development of human society. It can be said that this is a very meaningful work. Then, I have only minor suggestions to improve the paper.

1. At the end of the introduction, the particularity of Xinjiang is not enough, so it is suggested to improve it, which is more conducive to highlighting the necessity of taking Xinjiang as a case study.

2. Among the five dimensions that constitute the index system, the connotation of social integration is not clear, so it is suggested to further improve it.

3. In section 3.1.1 of the paper, the word order can be adjusted to put the sentences with conclusions and opinions before the article, so that readers can form an overall cognition of the overall situation of sustainable development of urbanization in Xinjiang from a macro perspective.

4. I suggest that some references should be added to the discussion section as the supporting basis for the views of this paper.

5. The overall length of the paper is long, which can be further simplified.

Author Response

Response to reviewers

Dear Editor and Reviewers:
Many thanks for the insightful comments and suggestions concerning our manuscript (ID: ijerph-2138243). We have considered your comments and have incorporated almost all your suggestions in the revised version of our paper. We also have a detailed response to each comment in the below. In the paper, the changes made are clearly highlighted using the "Track Changes" function. We feel that by revising our paper according to your constructive suggestions, the paper has been greatly improved.
The following are the answers and revisions.We have made in response to the reviewers' questions and suggestions on an item by item basis.

Reply to Referee 2
Specific Comments 1:
At the end of the introduction, the particularity of Xinjiang is not enough, so it is suggested to improve it, which is more conducive to highlighting the necessity of taking Xinjiang as a case study. 
Response to comment 1:
Thanks for the valuable suggestion. According to the comments, we have We further emphasize the special problems and challenges in Xinjiang’s urbanization process, as pointed out in the article, that Xinjiang's urbanization has both universality and particularity. Xinjiang's security environment, ecological environment, resource endowment, and carrying function of urbanization are all unique. Its urbanization must take national defense security and social stability into account, adapt to the ecological environment, and integrate into ethnic unity and regional culture. Please check the revised version in Line 148-152.

Specific Comments 2:
Among the five dimensions that constitute the index system, the connotation of social integration is not clear, so it is suggested to further improve it.
Response to comment 2:
Thanks for the valuable suggestion. According to the comment, we We have made a certain summary of the connotation of social integration, more clearly pointed out that the dimension of social integration covers many aspects that need to be paid attention to in the sustainable development process of Xinjiang's urbanization, including ethnic integration, urban-rural integration, industry-city integration, etc. Please check the revised version in Line 238-240

Specific Comments 3:
In section 3.1.1 of the paper, the word order can be adjusted to put the sentences with conclusions and opinions before the article, so that readers can form an overall cognition of the overall situation of sustainable development of urbanization in Xinjiang from a macro perspective.
Response to comment 3:
Thanks for the helpful suggestion. We have adjusted the word order and put the sentences with conclusions and opinions before the paragraph. Please check the revised version in Section 3.1.1.

Specific Comments 4:
I suggest that some references should be added to the discussion section as the supporting basis for the views of this paper.
Response to comment 4:
Thanks for the helpful suggestion. We have added a certain number of references in the discussion part as the supporting basis for the views of this paper. Please check the revised version.

Specific Comments 5:
The overall length of the paper is long, which can be further simplified.
Response to comment 5:
Thanks for the suggestion. We have made some deletions to some extent in order to ensure the simplicity of the paper. Please check the revised version.
